# HYDRA: Pruning Adversarially Robust Neural Networks

**Vikash Sehwag**✤, **Shiqi Wang**∗, **Prateek Mittal**✤, **Suman Jana**∗
✤Princeton University, USA     ∗Columbia University, USA

## Abstract

In safety-critical but computationally resource-constrained applications, deep learning faces two key challenges: lack of robustness against adversarial attacks and large neural network size (often millions of parameters). While the research community has extensively explored the use of robust training and network pruning *independently* to address one of these challenges, only a few recent works have studied them jointly. However, these works inherit a heuristic pruning strategy that was developed for benign training, which performs poorly when integrated with robust training techniques, including adversarial training and verifiable robust training. To overcome this challenge, we propose to make pruning techniques aware of the robust training objective and let the training objective guide the search for which connections to prune. We realize this insight by formulating the pruning objective as an empirical risk minimization problem which is solved efficiently using SGD. We demonstrate that our approach, titled HYDRA[1], achieves compressed networks with *state-of-the-art* benign and robust accuracy, *simultaneously*. We demonstrate the success of our approach across CIFAR-10, SVHN, and ImageNet dataset with four robust training techniques: iterative adversarial training, randomized smoothing, MixTrain, and CROWN-IBP. We also demonstrate the existence of highly robust sub-networks within non-robust networks. Our code and compressed networks are publicly available[2].

## 1   Introduction

How can we train deep neural networks (DNNs) that are robust against adversarial examples while minimizing the size of the neural network? In safety-critical and resource-constrained environments, both robustness and compactness are *simultaneously* necessary. However, existing work is limited in its ability to answer this question since it has largely addressed these challenges in *isolation*. For example, neural network pruning is an efficient approach to minimize the size of the neural networks. In parallel, robust training techniques can significantly improve the adversarial robustness of neural networks. However, improving adversarial robustness has been shown to require even larger neural networks [30, 49]. Thus it is even more critical

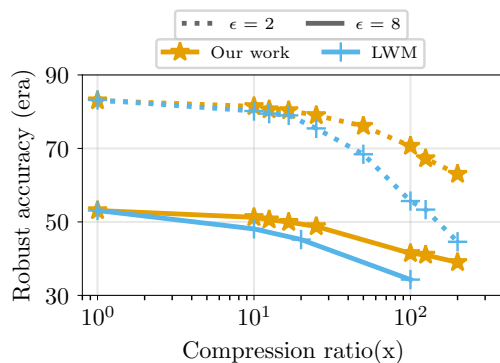

**Figure 1:** *Comparison of our proposed approach (★) and least-weight magnitude based pruning (+) for adversarial training ($l_\infty \leq \epsilon$) with VGG16 network and CIFAR-10 dataset. For both a weaker adversary (ε=2/255) or a stronger adversary (ε=8/255), our proposed technique leads to higher empirical robust accuracy where the gap increases with compression ratio.*

to ask *whether network pruning techniques can reduce the size of the network, i.e., number of connections, while preserving robustness?*

A gold standard for network pruning has been the approach of Han et al. [19], which prunes connections that have the lowest weight magnitude (LWM) under the assumption that they are the least useful. Sehwag et al. [34] demonstrated early success of LWM pruning with adversarially robust networks while Ye et al. [45] and Gui et al. [15] further improved its performance by integrating with alternating direction method of multipliers (ADMM) based optimization. These works inherit the heuristic assumption that connections with the least magnitude are also unimportant in the presence of robust training. While both LWM and ADMM based pruning techniques are highly successful with benign training [19, 50], they incur a huge performance degradation with adversarial training. Our design goal is to develop a pruning technique which achieves high performance and also generalizes to multiple types of robust training objectives including *verifiable* robustness [30, 49, 38, 48, 8].

Instead of inheriting a pruning heuristic and applying it to all robust training objectives, we argue that a better approach is to *make the pruning technique aware of the robust training objective itself*. We achieve this by formulating the pruning step, i.e., deciding which connections to prune, as an empirical risk minimization problem with a robust training objective, which can be solved efficiently using stochastic gradient descent (SGD). Our formulation is generalizable and can be integrated with multiple types of robust training objectives including verifiable robustness. Given a pre-trained network, we optimize the importance score [32] for each connection in the pruning step while keeping the fine-tuning step intact. Connections with the lowest importance scores are pruned away. We propose a scaled initialization of importance scores, which is a key driver behind the high benign and robust accuracy of our compressed networks.

Our proposed technique achieves much higher robust accuracy compared to LWM. Fig. 1 shows these results for adversarial training with both a weaker ($\epsilon$=2) and a stronger ($\epsilon$=8) adversary. With increasing pruning ratios, the gap between the robust accuracy achieved with both techniques further increases. Due to the accuracy-robustness trade-off in DNNs [49, 30], a rigorous comparison of pruning techniques should consider both benign and robust accuracy. We demonstrate that our compressed networks *simultaneously* achieve both state-of-the-art benign and robust accuracy.

Recently, Ramanujan et al. [32] demonstrated that there exist hidden sub-networks with high benign accuracy within randomly initialized networks. Using our pruning technique, we extend this observation to robust training, where we uncover *highly robust (both empirical and verifiable) sub-networks within non-robust networks*. In particular, within empirically robust networks that have no verifiable robustness, we found sub-networks with verified robust accuracy close to the state-of-the-art [33].

**Key contributions:** We make the following key contributions.

- We develop a novel pruning technique, which is aware of the robust training objective, by formulating it as an empirical risk minimization problem, which we solve efficiently with SGD. We show the generalizability of our formulation by considering multiple types of robust training objectives, including verifiable robustness. We employ an importance score based optimization technique with our proposed scaled initialization of importance scores, which is the key driver behind the success of our approach.

- We evaluate the proposed approach across four robust training objectives, namely iterative adversarial training [7, 30, 49], randomized smoothing [8, 7], MixTrain [38], and CROWN-IBP [48] on CIFAR-10, SVHN, and ImageNet dataset with multiple network architectures. Notably, at 99% connection pruning ratio, we achieve gains up to 3.2, 11.2, and 17.8 percentage points in robust accuracy, while simultaneously achieving state-of-the-art benign accuracy, compared to previous works [34, 45, 15] for ImageNet, CIFAR-10, and SVHN dataset, respectively.

- We also demonstrate the existence of highly robust sub-networks within non-robust or weakly robust networks. In particular, within empirically robust networks that have no verifiable robustness, we were able to find sub-networks with verified robust accuracy close to state-of-the-art.

## 2 Background and related work

**Robust training.** Robust training is one of the primary defenses against adversarial examples [5, 13, 6, 30, 3] where it can be divided into two categories: Adversarial training and verifiable robust training. The key objective of adversarial training is to minimize the training loss on adversarial

examples obtained with iterative adversarial attacks, such as projected gradient descent (PGD) [30] based attacks, under the following formulation.

$$\min_{\theta} \mathop{E}_{(x,y)\sim D} L_{adv}(\theta, x, y, \Omega), \qquad L_{adv}(\theta, x, y, \Omega) = L(\theta, \mathop{PGD}_{\delta \in \Omega}(x), y) \qquad (1)$$

Verifiable robust training provides provable robustness guarantees by minimizing a sound over-approximation to the worse-case loss $L_{ver}(\theta, x, y, \Omega)$ under a given perturbation budget. We focus on two state-of-the-art verifiable robust training approaches: (1) MixTrain [38] based on linear relaxations, and (2) CROWN-IBP [47] based on interval bound propagation (IBP). We also consider randomized smoothing [8, 26, 33, 24], which aims to provide certified robustness by leveraging network robustness against Gaussian noise.

**Neural network pruning.** Network pruning aims to compress neural networks by reducing the number of parameters to enhance efficiency in resource-constrained environments [19, 18, 27, 11, 25, 16, 29, 23]. One such highly successful approach is a three-step compression pipeline [19, 16]. It involves pre-training a network, pruning it, and later fine-tuning it. In the pruning step, we obtain a binary mask ($\hat{m}$), which determines which connections are most important. In the fine-tuning step, we only update the non-pruned connections to recover the performance. We refer the network obtained after fine-tuning as the *compressed network*. Note that both pruning and fine-tuning steps can be alternatively repeated to perform multi-step pruning [19], which incurs extra computational cost. In addition to this compression pipeline, network pruning can be performed with training, i.e, run-time pruning [28, 4] or before training [11, 25, 37]. We focus on pruning after training, in particular, LWM based pruning, since it still outperforms multiple other techniques (Table 1 in Lee et al. [25]) and is a long-standing gold standard for pruning techniques.

**Pruning with robust training.** Sehwag et al. [34] demonstrated that empirical adversarial robustness can be achieved with LWM based pruning heuristic. Ye et al. [45] and Gui et al. [15] further employed an alternating direction method of multipliers (ADMM) pruning framework [50], while still using LWM based pruning heuristic, to achieve better empirical robustness for compressed networks. We refer these previous works as Adv-LWM and Adv-ADMM respectively. In contrast, our work introduces an intellectually different direction as we let the robust training objective itself decide which connections to prune. Our compressed networks achieve both better accuracy and robustness than the previous works. In addition, our work is also the first 1) to study network pruning with verifiable robust training where we achieve heavily pruned networks with high verifiable robust accuracy, and 2) to demonstrate robust and compressed networks for the ImageNet dataset.

Some of these works [15, 42] also focused on other aspects of compression which are also applicable to our technique, such as quantization of weights. Another related line of research aims to use pruning itself to instill robustness against adversarial examples [10, 41, 17]. However, either these works are not successful at very high very pruning ratios [17, 41] (we focus on $\geq 90\%$ pruning ratios) or give a false sense of security as the robustness is diminished in the presence of an adaptive attacker [10, 2].

## 3   HYDRA: Our approach to network pruning with robust training

A central question in making robust networks compact is to decide *which connections to prune?* In LWM based pruning, irrespective of the training objective, connections with lowest weight magnitude are pruned away, with the assumption that those connections are the least useful. We argue that a better approach would be to perform an architecture search for a neural network with the desired pruning ratio that has the least drop in targeted accuracy metric compared to the pre-trained network.

We achieve this by formulating pruning as an empirical risk minimization (ERM) problem and integrating it with a robust training objective. Our formulation is generalizable where we show its integration with multiple empirical and verifiable robust training objectives, including adversarial training, MixTrain, CROWN-IBP, and randomized smoothing. We employ an importance score based optimization [32] approach to solve the ERM problem. However, we find that naive initialization of importance scores [20, 12, 32] brings little to no gain in the performance of compressed networks. We thus propose a *scaled initialization* of importance scores, and show that it enables our approach to simultaneously achieve state-of-the-art benign and robust accuracy at high pruning ratios. In addition, we also demonstrate the existence of hidden robust sub-networks within non-robust networks.

**Pruning as an empirical risk minimization problem (ERM) with adversarial loss objectives.**
To recover performance loss incurred with a pruning heuristic such as LWM, a standard approach is
to fine-tune the network. In contrast, we explicitly aim to reduce the degradation of performance in
the pruning step itself. We achieve this by integrating the robust training objective in the pruning
strategy itself by formulating it as the following learning problem.

$$\hat{m} = \underset{m \in \{0, 1\}^N}{argmin} \; \underset{(x,y) \sim \mathcal{D}}{E} [L_{pruning}(\theta_{pretrain} \odot m, x, y)] \qquad s.t. \; \|m\|_0 \leq k \qquad (2)$$

$\theta \odot m$ refers to the element-wise multiplication of mask ($m$) with the weight parameters ($\theta$). Predefined
pruning ratio of the network can be written as $\left(1 - \frac{k}{N}\right)$, where $k$ is the number of parameters we keep
after pruning and $N = |\theta_{pretrain}|$ is the total number of parameters in the pre-trained network. Our
formulation is generalizable and can be integrated with different types of robust training objectives
by selecting $L_{pruning}$ equal to $L_{adv}$ or $L_{ver}$ (Section 2). Since the distribution $\mathcal{D}$ is unknown, we
minimize the empirical loss over the training data using SGD. The generated pruning mask $\hat{m}$ is then
used in the fine-tuning step.

**Importance score based optimization.** It is challenging to directly optimize over the mask $m$ since
it is binary (either the weight parameter is pruned or not). Instead, we follow the importance scores
based optimization [32]. It assigns an importance score (floating-point) to each weight indicating
its importance to the predictions on all input samples and optimizes based on the score. While
making a prediction, it only selects the top-k weights with the highest magnitude of importance
scores. However, on the backward pass, it will update all scores with their gradients.

**Scaled-initialization.** We observe that the performance of the proposed pruning approach depends
heavily on the initialization of importance scores. At high pruning ratios, which we study in
this work, we observe slow and poor convergence of SGD with random initialization [20, 12] of
importance scores. To overcome this challenge, we propose a scaled initialization for importance
scores (Equation 3) where instead of random values, we initialize importance scores proportional
to pre-trained network weights. With scaled-initialization we thus give more importance to large
weights at the start and let the optimizer find a better set of pruned connections.

$$s_i^{(0)} \propto \frac{1}{max(|\theta_{pretrain,i}|)} \times \theta_{pretrain,i} \qquad (3)$$

where $\theta_{pretrain,i}$ is the weight corresponding to $i_{th}$ layer in the pre-trained network. We normalize
each layer weight to map it to [-1, 1] range. For the concrete scaling factor in Eq. 3, we use $\sqrt{\frac{6}{fan\text{-}in_i}}$,
motivated from He et al. [20], where *fan-in* is the product of the receptive field size and the number
of input channels. We provide additional ablation studies on choice of different scaling factors in
Appendix B.1. We summarize our pipeline to compress networks in Algorithm 1.

While our approach to solving the optimization problem in the pruning step is inspired by Ramanujan
et al. [32], our key objective is to focus on adversarially robust networks, which is different from their
work. In addition, as we demonstrate in section 4.1, without the proposed scaled initialization, solving
the optimization problem in the pruning step brings negligible gains. The objective in Ramanujan et
al. [32] is to find sub-networks with high *benign* accuracy, hidden in a *randomly initialized* network,
without the use of fine-tuning. Next, we present a more general formulation of their objective below.

**Imbalanced training objectives: Hidden robust sub-networks within non-robust networks.** To
optimize for a robustness metric in the compressed network, we use its corresponding loss function in
both pre-training and pruning. *But what if we use different loss functions in pre-training and pruning
steps (no fine-tuning)?* For example, if we select $L_{pruning} = L_{ver}$, where $L_{pretrain} = L_{benign}$, it will
search for verifiable robust sub-network within a benign, non-robust network. Using our pruning
approach, we uncover the existence of robust sub-networks within non-robust networks in Section 5.

# 4 Experiments

We conduct extensive experiments across three datasets, namely CIFAR-10, SVHN, and ImageNet.
We first establish strong baselines and then show that our method outperforms them significantly and
achieves state-of-the-art accuracy and robustness simultaneously for compressed networks.

**Setup.** We experiment with VGG-16 [36], Wide-ResNet-28-4 [46], CNN-small, and CNN-large [43] network architectures. The $l_\infty$ perturbation budget for adversarial training is 8/255 for CIFAR-10, SVHN and 4/255 for ImageNet. For verifiable robust training, we choose an $l_\infty$ perturbation budget of 2/255 in all experiments. These design choices are consistent with previous work [7, 38, 47]. We used PGD attacks with 50 steps and 10 restarts to measure *era*. We use state-of-the-art adversarial training approach from Carmon et al. [7] which supersedes earlier adversarial training techniques [30, 49, 22]. We present a detailed version of our experimental setup in appendix A.

## 4.1 Network pruning with HYDRA

We now demonstrate the success of our pruning technique in achieving highly compressed networks. In this subsection, we will focus on CIFAR-10 dataset, VGG-16 network, and adversarial training [7]. We present our detailed results later in Section 4.2 across multiple datasets, networks, pruning ratios, and robust training techniques. Our results (Table 1, 2 and Figure 1) demonstrate that the proposed method can achieve compression ratio as high as 100x while achieving much better benign and robust accuracy compared to other baselines.

**Metrics** We use following metrics to capture the performance of trained networks. 1) *Benign accuracy*: It is the percentage of correctly classified benign (i.e., non-modified) images. 2) *Empirical robust accuracy (era)*: It refers to the percentage of correctly classified adversarial examples generated with projected gradient descent based attacks. 3) *Verified robust accuracy (vra)*: Vra corresponds to verified robust accuracy, and **vra-m**, **vra-t**, and **vra-s** correspond to vra obtained from MixTrain, CROWN-IBP, and randomized smoothing, respectively. We refer pre-trained networks as PT.

In summary: 1) HYDRA improves both benign accuracy and robustness *simultaneously* over previous works (including Adv-LWM and Adv-ADMM), 2) HYDRA improves performance at multiple perturbations budgets (see Figure 1), 3) HYDRA's improvements over prior works increase with compression ratio, and 4) HYDRA generalizes as it achieves state-of-the-art performance across four different robust training techniques (Section 4.2).

---

**Algorithm 1** End-to-end compression pipeline.

**Inputs**: Neural network parameters ($\theta$), Loss objective: $L_{pretrain}, L_{prune}, L_{finetune}$, Importance scores ($s$), pruning ratio ($p$)

**Output**: Compressed network, i.e., $\theta_{finetune}$

Step 1: Pre-train the network.
$$\theta_{pretrain} = \underset{\theta}{argmin} \underset{(x,y)\sim\mathcal{D}}{E}[L_{pretrain}(\theta,x,y)]$$

Step 2: Initialize scores ($s$) for each layer.

$$s_i^{(0)} = \sqrt{\frac{6}{fan\text{-}in_i}} \times \frac{1}{max(|\theta_{pretrain,i}|)} \times \theta_{pretrain,i}$$

Step 3: Minimize pruning loss.
$$\hat{s} = \underset{s}{argmin} \underset{(x,y)\sim\mathcal{D}}{E}[L_{prune}(\theta_{pretrain},s,x,y)]$$

Step 4: Create binary pruning mask $\hat{m} = \mathbb{1}(|\hat{s}| > |\hat{s}|_k)$, $|\hat{s}|_k$: $k$th percentile of $|\hat{s}|$, $k = 100 - p$

Step 5: Finetune the non-pruned connections, starting from $\theta_{pretrain}$.
$$\theta_{finetune} = \underset{\theta}{argmin} \underset{(x,y)\sim\mathcal{D}}{E}[L_{finetune}(\theta \odot \hat{m}, x, y)]$$

---

**Table 1:** *Benign accuracy/*era *of compressed networks obtained with pruning from scratch, LWM, random initialization, and proposed pruning technique with scaled initialization. We use CIFAR-10 dataset and VGG16 network in this experiment.*

| Pruning ratio | PT | 90% | 95% | 99% |
|---|---|---|---|---|
| *Scaled-initialization* | | **80.5/49.5** | **78.9/48.7** | **73.2/41.7** |
| Scratch | | 74.7/45.6 | 71.5/42.3 | 34.4/24.6 |
| LWM [19] | 82.7 | 78.8/47.7 | 76.7/45.2 | 63.2/34.1 |
| Xavier-normal | /51.9 | 74.8/45.2 | 72.5/42.3 | 65.4/36.8 |
| Xavier-uniform | | 75.1/45.0 | 73.0/42.4 | 65.8/36.5 |
| Kaiming-normal | | 75.3/44.9 | 72.4/42.1 | 66.3/36.5 |
| Kaiming-uniform | | 75.0/44.8 | 73.3/42.5 | 66.1/36.4 |

**Table 2:** *Comparison of our approach with Adv-ADMM based pruning. We use CIFAR-10 dataset and VGG16 networks, iterative adversarial training from Madry et al. [30] for this experiment.*

| Pruning ratio | PT | 90% | 95% | 99% |
|---|---|---|---|---|
| Adv-ADMM | | 76.3/44.4 | 72.9/43.6 | 55.2/34.1 |
| HYDRA | 79.4/44.2 | **76.6/45.1** | **74.0/44.7** | **59.9/37.9** |
| $\Delta$ | | +0.3/+0.7 | +1.1/+1.1 | +4.7/+3.8 |

---

A key driver behind the success of HYDRA is the scaled initialization of importance scores in pruning. We found that widely used random initializations [20, 12], with either Gaussian or uniform distribution, perform even worse than training from scratch, our first baseline which we discuss below (Table 1). In contrast, architecture search with our scaled-initialization can significantly improve both benign accuracy and *era* of the compressed networks simultaneously, at all pruning ratios. To delve

deeper, we further compare the performance of each initialization technique at each epoch in the optimization for the pruning step (Figure 2 left). We find that with the proposed initialization, SGD converges faster and to a better pruned network, in comparison to widely used random initializations. It is because, with our initialization, SGD enjoys much higher magnitude gradients (using $\ell_2$ norm) throughout the optimization in the pruning step (Figure 2 right).

**Validating empirical robustness with stronger attacks.** To further determine that robustness in compressed networks in not arising from phenomenon such as gradient masking [31, 2], we evaluate them with much stronger PGD attacks (up to 100 restarts and 1000 attack steps) along with an ensemble of gradient-based and gradient-free attacks [9]. Our results confirm that the compressed networks show similar trend as non-compressed nets with these attacks (Appendix A.1). Note that *vra* already provides a lower bound of robustness against *all possible attacks* in the given threat model.

**Comparison with training from scratch.** If the objective is to achieve a compressed and robust network, a natural question is why not train on a compact network from scratch? However, we observe in Table 1 that it achieves poor performance. For example, at 99% pruning ratio, the compressed network has only 24.6% *era* which is 27.3 and 17.1 percentage points lower than the non-compressed network and our approach, respectively. We present a detailed analysis in Appendix C.1.

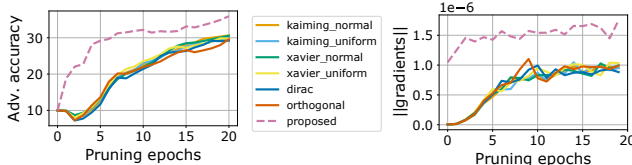

| Network | Adv-ADMM | Ours | Δ |
|---|---|---|---|
| ResNet-18 | 58.7/36.1 | 69.0/41.6 | +10.3/+5.5 |
| ResNet-34 | 68.8/41.5 | 71.8/44.4 | +3.0/+2.9 |
| ResNet-50 | 69.1/42.2 | 73.9/45.3 | +4.8/+3.1 |
| WRN-28-2 | 48.3/30.9 | 54.2/34.1 | +5.9/+3.2 |
| GoogleNet | 53.4/33.8 | 66.7/40.1 | +13.3/+6.3 |
| MobileNet-v2 | 10.0/10.0 | 39.7/26.4 | +29.7/+16.4 |

**Figure 2:** *We compare our proposed initialization with six other widely used initializations. With proposed initialization in the pruning step, SGD converges faster and to a better architecture (left), since it enjoys higher magnitude gradients throughout (right). (CIFAR10, 99% pruning)*

**Table 3:** *Comparing test accuracy/robustness (era) with Adv-ADMM (CIFAR10 dataset, 99% pruning). Our approach outperforms Adv-ADMM across all network architectures.*

**Comparison with Adv-LWM based robust pruning.** LWM based pruning with robust training (following Sehwag et al. [34]) is able to partially improve the robustness of compressed networks compared to training from scratch. At 99% pruning ratio, it improves the *era* to 34.1% but this is still 17.8 and 7.6 percentage points lower than a non-compressed network and our proposed approach, respectively. We observe similar gaps when varying adversarial strength in adversarial training (Figure 1). Furthermore, our approach also achieves up to 10 percentage points higher benign accuracy compared to Adv-LWM. Note that our method also outperforms LWM based pruning with benign training (Appendix C.6). We use only 20 epochs in the pruning step (with 100 epochs in both pre-training and fine-tuning), thus incurring only $1.1\times$ the computational overhead over Adv-LWM.

**Comparison with Adv-ADMM based robust pruning.** Finally, we compare our approach with ADMM based robust pruning [45, 15] in Table 2. Note that Ye et al. [45] have reported results with a former adversarial training technique [30] while we use the state-of-the-art techniques [49, 7]. Thus for a fair comparison, we use the exact same adversarial training technique, network architecture, and pre-trained network checkpoints as their work. Our approach outperforms ADMM based pruning at every pruning ratio and achieves up to 4.7 and 3.8 percentage point improvement in benign accuracy and *era*, respectively (Table 2). Adv-ADMM uses 100 epochs in pruning (compared to 20 epochs in our work), making it $5\times$ and $1.36\times$ more time consuming than our approach in the pruning step and overall, respectively. We also provide a comparison along six more recent network architectures (Table 3). Our method achieves better accuracy and robustness, simultaneously, across all of them. Furthermore, while Adv-ADMM fails to even converge for MobileNet, which was already designed to be a highly compact network, we achieve non-trivial performance. In addition, while Adv-ADMM has been shown to work with only adversarial training, our approach also generalizes to multiple verifiable robust training techniques.

**Ablation studies.** First, we vary the amount of data used in solving ERM in pruning step. Though a small number of images do not help much, the transition happens around 10% of the training data (5k images on CIFAR-10) after which an increasing amount of data helps in significantly improving the *era* (Appendix B.2). Next, we vary the number of epochs in the pruning step from one to a hundred. We observe that even a small number of pruning epochs, such as five, are sufficient to achieve large gains in *era* and further gains start diminishing as we increase the number of epochs (Appendix B.3).

**Table 4:** *Experimental results (benign/robust accuracy) for empirical test accuracy (*era*) and verifiable robust accuracy based on MixTrain (*vra-m*), randomized smoothing (*vra-s*), and CROWN-IBP (*vra-t*).*

**(a)** *Adversarial training (*era*)*

| Architecture | | VGG-16 | | | WRN-28-4 | | |
|---|---|---|---|---|---|---|---|
| Method | | Adv-LWM | HYDRA | Δ | Adv-LWM | HYDRA | Δ |
| CIFAR-10 | PT | | 82.7/51.9 | | | 85.6/57.2 | |
| | 90% | 78.8/47.7 | **80.5/49.5** | +0.7/+1.8 | 82.8/53.8 | **83.7/55.2** | +0.9/+1.4 |
| | 95% | 76.7/45.2 | **78.9/48.7** | +2.2/+3.5 | 79.3/48.8 | **82.7/54.2** | +3.4/+5.4 |
| | 99% | 63.2/34.1 | **73.2/41.7** | +10.0/+7.6 | 66.6/36.1 | **75.6/47.3** | +9.0/+11.2 |
| SVHN | PT | | 90.5/53.5 | | | 93.5/60.1 | |
| | 90% | 89.2/51.5 | 89.2/**52.4** | 0/+0.9 | 92.3/59.4 | **94.4/62.8** | +2.1/+3.4 |
| | 95% | 84.9/50.4 | **85.5/51.7** | +0.6/+1.3 | 90.4/53.4 | **93.0/59.8** | +2.6/+6.4 |
| | 99% | 50.4/29.0 | **84.3/46.8** | +33.9/+17.8 | **82.8**/45.3 | 82.2/**52.4** | - 0.6/+7.1 |

**(b)** *Randomized smoothing (*vra-s*)*

| Architecture | | VGG-16 | | | WRN-28-4 | | |
|---|---|---|---|---|---|---|---|
| Method | | Adv-LWM | HYDRA | Δ | Adv-LWM | HYDRA | Δ |
| CIFAR-10 | PT | | 82.1/61.1 | | | 85.7/63.3 | |
| | 90% | 82.3/59.6 | **83.4/60.7** | +1.1/+1.1 | 82.3/61.0 | **85.6/63.0** | +3.3/+2.0 |
| | 95% | 80.3/56.8 | **83.1/59.9** | +2.8/+3.1 | 80.3/59.9 | **84.5/62.5** | +4.2/+2.4 |
| | 99% | 65.1/44.1 | **77.1/54.4** | +12.0/+10.3 | 65.1/49.1 | **78.2/56.0** | +13.1/+6.9 |
| SVHN | PT | | 92.8/60.1 | | | 92.7/62.2 | |
| | 90% | 92.4/59.9 | **92.7**/59.9 | +0.3/0.0 | 92.4/62.2 | **92.8/62.3** | +0.4/+0.1 |
| | 95% | 92.2/**59.8** | **92.4**/59.3 | +0.2/- 0.6 | 92.2/61.4 | **93.1/62.0** | +0.9/+0.6 |
| | 99% | 87.5/51.9 | **91.4/58.6** | +3.9/+6.7 | 87.5/45.0 | **91.8/59.6** | +4.3/+14.6 |

**(c)** *CROWN-IBP (*vra-t*)*

| Architecture | | CNN-small | | | CNN-large | | |
|---|---|---|---|---|---|---|---|
| Method | | Adv-LWM | HYDRA | Δ | Adv-LWM | HYDRA | Δ |
| CIFAR-10 | PT | | 53.3/42.0 | | | 58.0/45.5 | |
| | 90% | 53.5/42.4 | **53.5/42.9** | +0.0/+0.5 | 58.9/46.9 | **59.1/47.0** | +0.2/+0.1 |
| | 95% | **49.7/40.3** | 49.5/40.0 | - 0.2/- 0.3 | 57.2/46.1 | **57.8/46.2** | +0.6/+0.1 |
| | 99% | 19.8/17.3 | **34.6/29.5** | +14.8/+12.2 | 42.9/34.6 | **47.7/39.4** | +4.8/+4.8 |
| SVHN | PT | | 59.9/40.8 | | | 68.5/47.1 | |
| | 90% | 59.1/40.3 | **60.4/40.6** | +1.3/+0.3 | 69.2/48.5 | **68.8/48.9** | - 0.4/+0.4 |
| | 95% | 49.4/34.8 | **53.0/36.7** | +3.6/+1.9 | 69.0/47.2 | **69.2/47.6** | +0.2/+0.4 |
| | 99% | 19.6/19.6 | 19.6/19.6 | 0.0/0.0 | 50.1/38.2 | **56.3/42.8** | +6.2/+4.6 |

**(d)** *MixTrain (*vra-m*)*

| Architecture | | CNN-small | | | CNN-large | | |
|---|---|---|---|---|---|---|---|
| Method | | Adv-LWM | HYDRA | Δ | Adv-LWM | HYDRA | Δ |
| CIFAR-10 | PT | | 62.5/46.8 | | | 63.8/47.7 | |
| | 90% | 46.9/35.3 | **54.8/41.0** | +7.9/+5.7 | 63.3/47.1 | **65.7/49.6** | +2.4/+2.5 |
| | 95% | 29.4/24.0 | **50.7/38.3** | +21.3/+14.3 | 50.6/39.3 | **60.2/45.3** | +9.6/+6.0 |
| | 99% | 10.0/10.0 | **27.0/24.9** | +17.0/+14.9 | 30.0/25.8 | **42.7/35.3** | +12.7/+9.5 |
| SVHN | PT | | 72.5/48.4 | | | 77.0/56.9 | |
| | 90% | 60.3/41.6 | **57.5/45.7** | - 2.8/+4.1 | 77.9/57.0 | **78.4/57.9** | +0.5/+0.9 |
| | 95% | 19.6/19.6 | **52.5/33.7** | +32.9/+14.1 | 19.6/19.6 | **74.8/53.7** | +55.2/+34.1 |
| | 99% | 19.6/19.6 | 19.6/19.6 | 0.0/0.0 | 19.6/19.6 | 19.6/19.6 | 0.0/0.0 |

## 4.2 Results across multiple datasets and robust training techniques

Table 4 presents the experimental results on CIFAR-10 and SVHN datasets across three pruning ratios, two network architectures, and four different robust training objectives. The key characteristics of the proposed pruning approach from these results are synthesized below:

**Improved robustness across datasets, architectures, and robust training objectives.** Across most experiments in Table 4, HYDRA achieves a significant improvement in robust accuracy with a mean and maximum improvement of 5.1 and 34.1 percentage points, respectively. Specifically, it achieves a mean improvement in robust accuracy by 5.6, 3.9, 2.0, 8.8 percentage points for adversarial training, randomized smoothing, CROWN-IBP, and MixTrain approach, respectively.

**Improved benign accuracy along with robustness.** Our approach not only improves robustness, but also the benign accuracy of pruned networks simultaneously across most experiments. Specifically, it achieves a mean improvement in benign accuracy by 5.4, 3.9, 2.6, 13.1 percentage points for adversarial training, randomized smoothing, CROWN-IBP, and MixTrain approach, respectively.

**Higher gains with an increase in pruning ratio.** At 99% pruning ratio, not only is our approach never worse than the baseline but it also achieves the highest gains in robust accuracy. For example, for VGG16 network with CIFAR-10 dataset at 99% pruning ratio, our approach is able to achieve 7.6 and 10.3 percentage points higher *era* and *vra-s*. These improvements are larger than the gains obtained at smaller pruning ratios. At very high pruning ratios for CROWN-IBP and MixTrain, the pruned networks with our approach are also more likely to converge.

**Help increase generalization for some cases.** Interestingly, we observe that our pruning approach can obtain robust accuracy even higher than pre-trained networks. For the SVHN dataset and WRN-28-4 network, we observe an increase by 2.7 and 0.1 percentage points for adversarial training and randomized smoothing, respectively at 90% pruning ratio. For verifiable training with CROWN-IBP, we observe improvement in *vra-t* from 0.9-1.8 percentage points for networks pruned at 90% ratio.

**Table 5:** *Era for ResNet50 network trained on ImageNet dataset with adversarial training for $\epsilon=4/255$.*

| Pruning ratio | PT | | 95% | | 99% | |
|---|---|---|---|---|---|---|
| | top-1 | top-5 | top-1 | top-5 | top-1 | top-5 |
| Adv-LWM | | | 45.0/19.6 | 70.2/43.3 | 24.8/9.8 | 47.8/24.4 |
| HYDRA | 60.2/32.0 | 82.4/61.1 | **47.1/21.4** | **72.2/46.6** | **31.5/13.0** | **56.2/31.2** |
| Δ | | | +2.1/+1.8 | +2.0/+3.3 | +6.7/+3.2 | +8.4/+6.8 |

Similar improvements are also observed for CNN-large with MixTrain. Note that the improvement mostly happens for WRN-28-4 and CNN-large architectures, where both networks achieve better robust accuracy than their counterparts. This suggests that there still exists a potential room for improving the generalization of these models with robust training. We present additional results in Appendix C.4.

**Performance on ImageNet dataset.** To assess the performance of pruning techniques on large-scale datasets, we experiment with the ImageNet dataset. Table 5 summarizes our results. Similar to smaller-scale datasets, our approach also outperforms LWM based pruning for the ImageNet dataset. In particular, at 99% pruning ratio, our approach improves the top-1 *era* by 3.2 percentage points, and the top-5 *era* by 6.8 percentage points.

# 5 Imbalanced training objectives: Hidden robust sub-networks within non-robust networks.

We have already demonstrated that the success of HYDRA stems from finding a set of connections which, when pruned, incurs least degradation in the pre-trained network robustness. *What if the pre-trained network is trained with a different objective than pruning*? To answer this question, we prune a pre-trained network with three different objectives (no fine-tuning), namely benign training, adversarial training, and randomized smoothing. These results are presented in Table 6 where the pruning ratio for each sub-network is 50% with VGG16 network and CIFAR-10 dataset.

**Table 6:** *Performance of sub-networks within pre-trained networks. Given a pre-trained network, we search for a sub-network optimized for one metric from benign accuracy,* era *($\epsilon$=8/255), or* vra-s *($\epsilon$=2/255).*

| Pre-training objective | | Targeted metric for each sub-network | | |
|---|---|---|---|---|
| | | benign accuracy | era | vra-s |
| Benign training | (benign accuracy = 95.0) | 95.0 | 43.5 | 53.0 |
| Adversarial training | (era = 51.9) | 94.1 | 51.4 | 63.6 |
| Randomized smoothing | (vra = 61.1) | 93.7 | 48.8 | 60.7 |

Our results show that *there exist highly robust sub-networks even within non-robust networks*. For example, we were able to find a sub-network with 43.5% *era* when the pre-trained network was trained with benign training and had 0% *era*. As a reference, the pre-trained network with adversarial training has 51.9% *era*.

Surprisingly, in networks pre-trained with adversarial training, we found sub-networks with really high verified robust accuracy from randomized smoothing. For example, when searched with randomized smoothing technique from Carmon et al [7], we found a sub-network with 63.6% robust accuracy on CIFAR10 dataset. [3] Note that the sub-network has a higher *vra-s* than 60.7%, which is achieved with a network pre-trained with randomized smoothing from Carmon et al. [7]. Under a similar setup, we also find a sub-network with 61.3% *vra-s* within an adversarially trained network on SVHN dataset. In comparison, a pre-trained network could only achieve 60.1% *vra-s*.

# 6 Delving deeper into network pruning and concluding remarks

**Visualizing which connections are being pruned.** We visualize the distributions of weights in pruned networks from our proposed approach and the Adv-LWM baseline in Figure 3. There are two key insights 1) Our search for a better pruned architecture is likely to find connections with very small magnitudes unnecessary and prunes them away. 2) *However, in contrast to the LWM heuristic, our approach does favor pruning some large-magnitude weights instead of smaller ones.* We present more detailed visualizations in Appendix D.

**Further compression after integration with quantization (Appendix C.7).** We found that our pruned networks (even at 99% pruning ratio) can be quantized by 8-bits while only incurring <0.5 percentage point decrease in both benign and robust accuracy. This brings another 4x compression factor in already heavily pruned (10x-100x) networks.

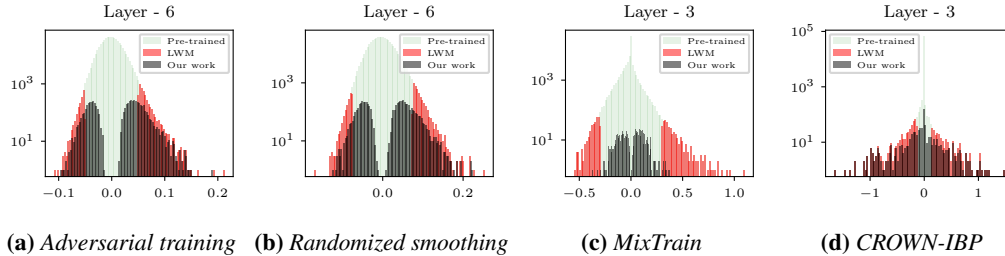

| (a) *Adversarial training* | (b) *Randomized smoothing* | (c) *MixTrain* | (d) *CROWN-IBP* |

**Figure 3:** *Comparison of the weights preserved by each pruning technique. In background, we display the histogram of weights for the pre-trained network. Then we show the weights preserved by each technique after pruning (without fine-tuning). Note that the proposed pruning technique tends to preserve small-magnitude weights as opposed to other large-magnitude weights preserved by LWM. We use 99% pruning ratio with VGG16 network in figure (a), (b) and with CNN-large networks in figure (c), (d), and train on CIFAR-10 dataset.*

**Multi-step pruning (Appendix C.8).** To reduce computational overhead, so far we only used a single pruning step. On ImageNet dataset, even when we use a multi-step (20-steps) Adv-LWM technique, our approach, which still uses a single pruning step, outperforms it by a large extent.

**Structured pruning (Appendix C.9).** Structured pruning, i.e., pruning filters instead of connections, has a much stronger impact on performance [27]. When pruning 50% filters with LWM technique, the *era* of VGG16 network decreases from 51% to 34.7% on CIFAR-10. Our approach achieves 38.0% *era* while also achieving 1.1 percentage point higher benign accuracy than Adv-LWM.

**Lower degradation in *era* for over-parameterized networks.** With 90% pruning for the over-parameterized WRN-28-10 network, we observe only 0.3 percentage point degradation in *era*, which is significantly lower than 1.4 percentage point degradation incurred for a smaller WRN-28-4 network.

### 6.1 Concluding Remarks

In this work, we study the interplay between neural network pruning and robust training objectives. We argue for integrating the robust training objective in the pruning technique itself by formulating pruning as an optimization problem and achieve state-of-the-art benign and robust accuracy, simultaneously, across different datasets, network architectures, and robust training techniques. An open research question is to further close the performance gap between non-pruned and pruned networks.

## Broader Impact

Our work provides an important capability for deploying machine learning in safety critical and resource constrained environments. Our compressed networks provide a pathway for higher efficiency in terms of inference latency, energy consumption, and storage. On the other hand, these networks provide robustness against adversarial examples, including verified robustness properties, mitigating test-time attacks on critical ML services and applications. Recent work has leveraged adversarial examples against neural networks for positive societal applications, such as pushing back against large-scale facial recognition and surveillance. The development of robust networks may hinder such societal applications. Nevertheless, it is important to understand the limits and capabilities of compressed networks in safety critical environments, as failure to develop robust systems can also have catastrophic consequences. Our approach does not leverage any biases in data.

## Acknowledgements

This work was supported in part by the National Science Foundation under grants CNS-1553437, CNS-1704105, by Qualcomm Innovation Fellowship, by the Army Research Office Young Investigator Prize, by Schmidt DataX Fund, by Princeton E-ffiliates Partnership, by Army Research Laboratory (ARL) Army Artificial Intelligence Institute (A2I2), by Facebook Systems for ML award, by a Google Faculty Fellowship, by a Capital One Research Grant, by an ARL Young Investigator Award, by the Office of Naval Research Young Investigator Award, by a J.P. Morgan Faculty Award, and by two NSF CAREER Awards.

## Footnotes

[1]a small organism with high resiliency and biological immortality due to regenerative abilities.

[2]https://github.com/inspire-group/compactness-robustness

[3]When searched with improved technique from Salman et al. [33], we find a even better sub-network with 64.3% *vra-s*.

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
