[Supplementary Material]

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

. For each dataset, we pre-train the networks with a learning rate of 0.1. We perform 100 training epochs for CIFAR-10, SVHN and 90 epochs for ImageNet. In the pruning step, we perform 20 epochs for CIFAR-10, SVHN and 90 epochs for ImageNet. We experiment with VGG-16 [36], Wide-ResNet-28-4 [46], CNN-small, and CNN-large [43] network architectures. Since both MixTrain and CROWN-IBP methods only work with small scale networks (without batch-normalization), we use only CNN-large and CNN-small for them. We split the training set into a 90/10 ratio for training and validation for tuning the hyperparameters. Once hyperparameters are fixed, we use all training images to report the final results.

**Adversarial training:** We use the state-of-the-art iterative adversarial training setup (based on PGD) with $l_\infty$ adversarial perturbations on CIFAR-10 and SVHN dataset. The maximum perturbation budget, the number of steps, and perturbations at each step are selected as 8, 10, and 2 respectively. In particular, for CIFAR-10, we follow the robust semi-supervised training approach from Carmon et al. [7], where it used 500k additional pseudo-labeled images from the TinyImages dataset. For ImageNet, we train using the free adversarial training approach with 4 replays and perturbation budget of 4 [35]. We evaluate the robustness of trained networks against a stronger attack, where we use 50 iterations for the PGD attack with the same maximum perturbation budget and step size.

**Provable robust training**: We evaluate our pruning strategy under three different provable robust training settings. We choose an $l_\infty$ perturbation budget of 2/255 in all experiments. These design choices are consistent with previous work [7, 38, 47].

- *MixTrain*: We use the best training setup reported in Wang et al. [38] for both CIFAR-10 and SVHN. In specific, we use sampling number $k'$ as 5 and 1 for CNN-small and CNN-large. We select $\alpha = 0.8$ to balance between regular loss and verifiable robust loss. The trained networks are evaluated with symbolic interval analysis [40, 39] to match the results in Wang et al. [38].

- *CROWN-IBP*: We follow the standard setting in Zhang et al. [47] for CROWN-IBP. We set the $\epsilon$ scheduling length to be 60 epochs (gradually increase training $\epsilon$ from 0 to the target one), during which we gradually decrease the portion of verifiable robust loss obtained by CROWN-IBP while increasing the portion obtained by IBP for each training batch. For the rest of the epochs after the scheduling epochs, only IBP contributes to the verifiable robust loss. We use IBP to evaluate the trained networks.

- *Randomized smoothing*: We train the network using the stability training for CIFAR-10 and SVHN dataset (similar to Carmon et al. [7]). We calculated the certified robustness with $N_0 = 100$, $N = 10^4$, noise variance ($\sigma$=0.25), and $\alpha = 10^{-3}$. We choose an $l_2$ budget of 110/255 which gives an upper bound on robustness against an $l_\infty$ budget of 2/255 for CIFAR-10 and SVHN dataset.

**Table 7:** *All neural network architectures, with their number of parameters, used in this work.*

| Name | Architecture | Parameters |
|---|---|---|
| VGG4 | conv 64 → conv 64 → conv 128 → conv 128 → fc 256 → fc 256 → fc 10 | 0.46m |
| VGG16 | conv 64 → conv 64 → conv 128 → conv 128 → conv 256 → conv 256 → conv 256 → conv 512 → conv 512 → conv 512 → conv 512 → conv 512 → conv 512 → fc 256 → fc 256 → fc 10 | 15.30m |
| CNN-small | conv 16 → conv 32 → fc 100 → fc 10 | 0.21m |
| CNN-large | conv 32 → conv 32 → conv 64 → conv 64 → fc 512 → fc 512 → fc 10 | 2.46m |
| WideResNet-28-4 | Proposed architecture from Zagoruyko et al.[46] | 6.11m |
| ResNet50 | Proposed architecture form He et al. [21] | 25.50m |

**Pruning and fine-tuning:** Except for learning rate and the number of epochs, pruning and fine-tuning have similar training parameters as pre-training. We choose the number of epochs as 20 in all experiments (if not specified). Similar to pre-training, for pruning we choose learning of 0.1 with cosine decay. Often when this learning rate is too high (in particular for MixTrain and CROWN-IBP), we report results with the learning of 0.001 for the pruning step. Fine-tuning is done with a learning rate of 0.01 and cosine decay. To make sure that the algorithm does not largely prune fully connected layers that have most parameters, we constrain it to prune each layer by equal ratio.

## A.1  Validating robustness against stronger attacks

Iterative adversarial training [30] has long withstood its performance against attacks of varied strength [2]. It is natural to ask whether our compressed networks bears the same strength. To evaluate it, we measure the robustness of our compressed network against stronger adversarial attacks.

**Increasing attack steps and the number of restarts.** With increasing step-size, i.e, enabling adversary to search for stronger adversarial examples, we choose the perturbation budget for each step with the $\frac{2.5*\epsilon}{steps}$ rule suggested by Madry et al. [30]. Figure 4 shows the results for networks trained on CIFAR-10 datasets. It shows that gains in adversarial attack strength saturate after a certain number of attacks steps since the robust accuracy

stops decreasing significantly. Similarly, with 100 random restarts for VGG-16 at 95% pruning ratio, we observe only a 0.6 percentage point decrease in *era* compared to the baseline. Note that the pre-trained network also incurs an additional 0.7 percentage point degradation in *era* with 100 random restarts, suggesting that compressed networks behave similarly to pre-trained, i.e., non-compressed, networks under stronger adversarial attacks. We use 50 attack steps with 10 restarts for all adversarial attacks in our evaluation.

**Evaluation with auto-attack [9].** With auto-attack, which is an ensemble of gradient-based and gradient-free attacks, we observe similar trend for compressed networks compared to non-compressed, i.e., pre-trained networks. For example, the *era* of pre-trained VGG-16 with auto-attacks is 48.3% (*3.6* percentage points lower than *era* with PGD-50 attack). In contrast, *era* of a 95% pruned VGG-16 network is 44.8, which is again only 3.7 percentage points lower than PGD-50 attack. In comparison to the PGD-50 baseline, the decrease in *era* with auto-attack is comparable for pre-trained, i.e., non-compressed, and pruned networks.

**Figure 4:** *Empirical adversarial accuracy (era) of compressed networks with increasing number of steps in projected gradient descent (PGD) based attack. Beyond a certain number steps,* era *is largely constant with increase in steps. Results are reported for compressed networks up to 99% pruning ratio with CIFAR-10 dataset.*

## A.2   Network architectures

Table 7 contains the architecture and parameters details of the neural networks used in this work. For WideResNet-28-4 and ResNet-50, we use the original architectures proposed in Zagoruyko et al. [46] and He et al. [21], respectively. CNN-large and CNN-small are similar to architectures used in Wong et al. [43]. VGG4 and VGG16 are the the variants of original VGG architecture [36].

## A.3   Comparison with Ramanujan et al. [32]

While our approach to solving the optimization problem in the pruning step is inspired by Ramanujan et al. [32], we note that the goals of the two works have several significant differences. Their work aims to find sub-networks with high *benign* accuracy, hidden in a *randomly initialized* network, without the use of fine-tuning. In contrast, (1) we focus on multiple types of robust training objectives, including verifiably robust training, (2) we employ pre-trained networks in our pruning approach, as opposed to randomly initialized networks, and (3) we argue for further fine-tuning of pruned networks resulted from the optimization problem to further boost performance. We further employ an additional scaled-initialization mechanism which is the key driver of the success of our pruning technique. In contrast to their work which searches for sub-networks close to 50% pruning ratio, our goal is to find highly compressed networks (up to 99% pruning ratio).

# B   Further details on ablation studies

In this section, we further discuss the ablation studies for the pruning step in detail.

## B.1   Choice of scaling factor in importance scores initialization

Recall that we use $s_i^{(0)} = \gamma \times \sqrt{\frac{6}{fan\text{-}in_i}} \times \frac{1}{max(|\theta_{pretrain,i}|)} \times \theta_{pretrain,i}$ to initialize the importance scores in each layer, where $\gamma$ is the scaling factor. We use $\sqrt{\frac{k}{fan-in}}$, with $k = 6$, as the scaling factor. Note the our choice of k is also motivated by an earlier work from He et al. [20]. We also provide an ablation study with different values of k in Table 8, where measure performance of each scaling factor after the pruning step, i.e., no fine-tuning. First it demonstrate that the performance without a scaling factor, i.e., $\gamma$=1, is much worse. Next it validate our choice of $k = 6$, as it outperforms other choice for $k$.

## B.2   How much data is needed for supervision in pruning?

We vary the number of samples used from ten to all training images in the dataset for solving the ERM in the pruning step for CIFAR-10 dataset at a 99% pruning ratio. Fig. 5 shows there results. Data corresponding to zero samples refers to the least weight-magnitude based heuristic as it is used to initialize the pruning step. As the amount of data (number of samples) used in the pruning step increases, the robustness of the pruned network after fine-tuning also increases. For CIFAR-10, a small number of images doesn't help much in finding a better

**Table 8:** *Ablation over different values of k in the choice of scaling factor for the proposed initialization of importance scores. We focus on the pruning step, i.e., no fine-tuning, for a VGG-16 network at 99% pruning ratio. We use k=6 in our experiments.*

| $k$ | No-scaling | 2 | 4 | 6 | 8 | 10 | 12 | 14 |
|---|---|---|---|---|---|---|---|---|
| Benign accuracy | 57.9 | 62.4 | 64.5 | **66.8** | 66.7 | 66.4 | 65.0 | 66.1 |
| *era* | 31.6 | 33.9 | 35.7 | **35.8** | 34.7 | 35.7 | 33.2 | 35.2 |

pruned network. However, the transition happens around 10% of the training data (5k images for CIFAR-10) after which an increasing amount of data helps in significantly improving the *era*.

### B.3  Number of training epochs for pruning.

We vary the number for epochs used to solve the *ERM* problem for the pruning step from one to hundred. For each selection, the learning rate scheduler is cosine annealing with a starting learning rate of 0.1. Fig. 6 shows these results where we can see that an increase in the number of epochs leads to a network with higher *era* after fine-tuning. Data corresponding to zero epochs refers to the least weight-magnitude based heuristic since it is used to initialize the pruning step. We can see that even a small number of pruning epochs are sufficient to achieve large gains in $era$ and the gains start diminishing as we increase the number of epochs.

**Figure 5:** Era *of compressed networks with varying number of samples used in the pruning step at 99% pruning ratio for a VGG16 network and CIFAR-10 dataset.*

**Figure 6:** Era *of compressed networks with varying number of epochs used in the pruning step with VGG16 network and CIFAR-10 dataset.*

## C  Additional experimental Results

In this section, we first study the impact of sparsity in the network in the presence of benign and robust training. Next, we present the limitation of least weight magnitude pruning in the presence of robust training and discuss the choice of this heuristic as a baseline. After that, we study the improvement in the generalization of some networks after the proposed pruning technique. Next, we provide additional visualization on comparison of both techniques across the end-to-end compression pipeline. After that, we demonstrate the success of the proposed pruning technique with benign training. Finally we present integration of pruning technique with quantization, multi-step pruning, and structured pruning.

### C.1  Sparsity hurts more with robust training.

We first study the impact of sparsity in the presence of benign training and adversarial training. Fig. 7 shows these results, where we train multiple networks from scratch with different sparsity ratio and report the fractional decrease in performance compared to the non-sparse network trained from scratch. For each training objective (adversarial training or benign training) and sparsity ratio, we train an individual VGG4 network. These results show that robustness decreases at a faster rate compared to clean accuracy with increasing sparsity. Consider robust training against a stronger adversary ($\epsilon$=8), where at 75% sparsity ratio, the *era* reduced to a fraction of 0.74 of the non-sparse network. The fractional decrease in test accuracy for a similar setup is only 0.92. Even

defending against a weaker adversary ($\epsilon$=2), robust accuracy is hard to achieve in the presence of sparsity. The fractional decrease in *era* is .79 against this weaker adversary at 75% sparsity level. With the increasing size of the baseline network, such as VGG16, WideResNet-28-4 size, the rate of degradation of robustness with sparsity decreases but it still decays faster than the test accuracy.

This observation is closely related to the previously reported relationship between adversarial training and the size of neural networks [30, 44]. In particular, Madry et al. [30] demonstrated that increasing the width of the network improves robust accuracy to a large extent. We complement these observations by highlighting that further reducing the number of parameters (before training) reduces the robustness at a much higher rate.

**Figure 7:** *Compression hurts more in presence of adversarial training. We plot the fraction decrease in accuracy (+) for networks trained with benign training and robustness for different networks trained with adversarial training against varying adversarial strength.*

**Figure 8:** *Comparison of LWM and proposed pruning technique across varying adversarial perturbation budget ($\epsilon$) in adversarial training for VGG16 network on CIFAR-10 dataset.*

## C.2 Combining network pruning with robust training

We can further integrate the network pruning pipeline with robust training by updating the loss objective. For example, to achieve an empirically robust network, we can pretrain and fine-tuning a network with adversarial training (selecting $L_{pt} = L_f = L_{adv}$). Similarly, for other robust training mechanisms, we can use their respective loss functions. Next, we discuss the limited performance of least weight magnitude (LWM) based pruning.

**Limitation of least weight magnitude based heuristic.** Though pruning with least weight magnitude based heuristic brings some gains in improving the robust accuracy of the network, there still exists a large room for improvement. For example, at a 99% pruning ratio for a VGG16 network, it still incurs a decrease in *era* by 17.6 percentage points compared to the non-pruned i.e., pre-trained network. We also observe a non-linear drop in performance with increasing adversarial strength in adversarial training. Consider Fig. 8, where we report the performance of the pre-trained networks along with the compressed network (at 99% pruning ratio) from the pruning pipeline for different adversarial perturbation budgets in adversarial training. Against a weaker adversary, where the pre-trained network is highly robust, weight-based pruning heuristic struggles to achieve high robustness after compression. At smaller perturbation budgets, this gap increases further with the increase in adversarial strength.

## C.3 Why focus on pruning and fine-tuning based compression pipeline

We focus on pruning and fine-tuning approach because it achieves the best results among all three pruning strategies namely pruning before training, run-time pruning, and pruning after training i.e., pruning and fine-tuning. This is because the other approaches are constrained and tend to do pruning in a less flexible manner or with incomplete information. On the other hand, despite the simplicity, pruning and fine-tuning based on least weight magnitude [19] can itself achieve highly competitive results [29]. With similar motivation, we integrate this approach with robust training and select it as the baseline. This simplicity also allows us to integrate different training objectives, such as adversarial training and verifiable robust training.

## C.4 Increase in generalization with pruning

For verifiable training with CROWN-IBP, we observe improvement in generalization across all experiments ranging from 0.9-1.5 percentage points. Note that both proposed and baseline techniques can improve the generalization. This further highlights how network pruning itself can be used to improve the generalization of

verified training approaches. Table 9 summarizes these results for proposed pruning methods where we observe improvement in robust accuracy after pruning at multiple pruning ratios.

## C.5 Additional comparisons across end-to-end pruning pipeline

In figure 9, we present additional comparisons of LWM and proposed pruning approach across the end-to-end compression pipeline. Though both approaches use the identical pre-trained network, the proposed approach searches for a better pruning architecture in the pruning steps itself. Fine-tuning further improves the performance of these networks. For the WRN-28-4 network on the SVHN dataset, we also observe that the fine-tuning step decreases the perfor-

**Table 9:** *Verified robust accuracy with CROWN-IBP with the proposed pruning methods for CNN-large network and CIFAR-10 dataset.*

| Pruning ratio | 0 | 10 | 30 | 50 | 70 | 80 | 90 |
|---|---|---|---|---|---|---|---|
| *vra-t* | 45.5 | 46.1 | 46.0 | 45.9 | 45.9 | 46.0 | 46.1 |

mance to some extent for the proposed approach. We hypothesize that this behavior could be due to an imbalance in the learning rate at the end of the pruning step and the start of the fine-tuning step. With further-hyperparameter tuning, our approach can achieve higher gains for this network. However, for an impartial comparison with baseline, we avoid excessive tuning of hyperparameters for the proposed approach and use a single set of hyperparameters across all networks. The results are reported on a randomly partitioned validation of the CIFAR-10 and SVHN dataset at a 99% pruning ratio.

**(a)** *CIFAR-10, VGG16*

**(b)** *CIFAR-10, WRN-28-4*

**(c)** *SVHN, VGG16*

**(d)** *SVHN, WRN-28-4*

**Figure 9:** *Comparison of proposed pruning approach with least weight magnitude (LWM) based pruning at 99% pruning ratio for robust training with iterative adversarial training.*

## C.6 Performance with benign training

In this work, we have largely focused on demonstrating the success of the proposed pruning approach with multiple robust training objectives. However, it is natural to ask whether the proposed approach also has the same advantage with benign training i.e., in the absence of an adversary. We compare the performance of LWM and our approach for VGG16 and WRN-28-4 across CIFAR-10 and SVHN dataset in Table 10. Similar to robust training, our approach is also successful with benign training where it outperforms LWM based pruning in all experiments. In particular, even at a 99% pruning ratio, the proposed approach can maintain the accuracy within 1.2 percentage points for the SVHN dataset.

**Table 10:** *Performance of LWM and proposed pruning technique for benign training.*

| Architecture | | VGG16 | | | WRN-28-4 | | |
|---|---|---|---|---|---|---|---|
| Pruning ratio | | 0% | 95% | 99% | 0% | 95% | 99% |
| CIFAR-10 | LWM | 95.1±0.1 | 93.2±0.1 | 86.1±0.1 | 95.8±0.2 | 94.9±0.2 | 89.2±0.2 |
| | HYDRA | | **94.6±0.1** | **90.4±0.2** | | **95.5±0.2** | **91.2±0.2** |
| SVHN | LWM | 95.9±0.1 | 95.5±0.1 | 93.6±0.1 | 96.4±0.1 | 96.1±0.1 | 93.9±0.1 |
| | HYDRA | | **95.6±0.2** | **95.2±0.1** | | **96.3±0.1** | **95.2±0.2** |

## C.7 Further compression after integration with quantization

We also observe that our pruned networks can be easily quantized up to 8-bits (additional $4\times$ compression) without leading to significant degradation of accuracy or robustness. We report these results in Table 11 for the VGG-16 network with CIFAR-10 dataset and 99% pruning ratio. It shows that the accuracy of even 99% pruned networks doesn't degrade beyond 0.4 percentage points up to 8-bits. We observe similar accuracy as original networks for up to 12-bits quantization. Note that the non-pruned network also incurs similar degradation in performance. Since the quantized networks have discontinuous gradients, thus not amenable to PGD attacks, we use transferability and black-box based attacks to measure robustness. For transfer-based attacks, we transfer adversarial examples from the original 32-bit width network. For a black-box attack, we use the Square attack [1]. While both attacks are only surrogate of PGD attacks, they do show a similar trend for both non-pruned and pruned networks at lower bit-widths.

**Table 11:** *Performance with up to 6-bit quantization for both non-pruned, i.e., pre-trained, and 99% pruned VGG-16 network using our technique on CIFAR-10 dataset.*

| Bits | Non-pruned | | | 99% pruned | | |
|---|---|---|---|---|---|---|
| | Benign accuracy | Robust accuracy | | Benign accuracy | Robust accuracy | |
| | | Transfer-based | Gradient-free | | Transfer-based | Gradient-free |
| 12 | 82.7 | 60.8 | 64.1 | 73.1 | 49.4 | 52.4 |
| 8 | 82.5 | 62.1 | 71.7 | 72.7 | 51.0 | 61.2 |
| 6 | 81.2 | 61.0 | 75.4 | 72.5 | 50.6 | 65.4 |

## C.8 Detailed results on multi-step pruning

We compare the performance of proposed technique (single-step) with multi-step pruning using LWM, and summarize the top-1 and top-5 *era* obtained by each pruning strategy in Table 12. Though multi-step pruning can increase the performance of LWM, our approach still outperforms it by a large extent. For example, at 95% pruning ratio, multi-step pruning increases the *era* by 0.3 percentage points but it is still 1.5 percentage points lower than our proposed approach. Note that the performance of our proposed techniques can also be further increased with a multi-step approach, which however will incur additional computational overhead.

## C.9 Structured pruning

We present our results with structured pruning, i.e., filter pruning, in Table 13 for a VGG-16 network with adversarially training on CIFAR-10 dataset. Our pruning approach outperforms Adv-LWM baseline for structured pruning too, where we achieve up to 1.1 and 3.3 percentage point increase in benign accuracy and *era* respectively.

**Table 12:** *Comparisons of top-1,5 era obtained by single step LWM, multi-step LWM, and proposed approach on ImageNet dataset.*

| Pruning ratio | | 95% | | 99% | |
|---|---|---|---|---|---|
| | | top-1 | top-5 | top-1 | top-5 |
| Adv-LWM | single step | 19.6 | 43.3 | 9.8 | 24.4 |
| Adv-LWM | multi-step | 19.9 | 45.2 | 8.4 | 23.2 |
| HYDRA | single-step | **21.4** | **46.6** | **13.0** | **31.2** |

**Table 13:** *Benign-accuracy/era for structured pruning on a VGG-16 networks and CIFAR-10 dataset.*

| Pruning ratio | 0 | 50 | 90 |
|---|---|---|---|
| Adv-LWM | 0/0 | 51.8/34.7 | 17.9/16.4 |
| HYDRA | | 52.9/38.0 | 18.3/16.7 |
| Δ | | +1.1/+3.3 | +0.4/+0.3 |

# D  Visualization of pruned weights

Recall that for each of the learning objectives, the SGD in the pruning step starts from the same solution obtained from the least weight-magnitude based pruning due to our scaled-initialization. However, with each epoch, we observe that SGD pruned *certain* connections with large magnitude as opposed to connections with smaller magnitude. Fig. 10, 11 shows the results after 20 epochs where a significant number of connections with smaller magnitude are not pruned (in contrast to the LWM approach). Another intriguing observation is that even the SGD based solver finds connections with very small magnitudes unnecessary and prunes them away. This phenomenon is particularly visible for adversarial training and randomized smoothing where a significant number of connections with very small magnitude are also pruned away by the solver. This phenomenon also exists for MixTrain and CROWN-IBP but the fraction of such connections is very small and thus not clearly visible in the visualizations. One reason behind this could be that both of these learning objectives are biased towards learning connections with smaller magnitudes [14].

Fig. 10, 11 present the visualization for pruned connection for adversarial training and randomized smoothing for each layer in the VGG16 network for CIFAR-10 dataset. Fig. 12, 13 presents similar visualization for MixTrain and CROWN-IBP for CNN-large networks.

**Figure 10:** *Histogram of weights pruned by the baseline and proposed technique for adversarial training.*

**Figure 11:** *Histogram of weights pruned by the baseline and proposed technique for randomized smoothing based training.*

**Figure 12:** *Histogram of weights pruned by the baseline and proposed technique for MixTrain.*

**Figure 13:** *Histogram of weights pruned by the baseline and proposed technique for CROWN-IBP.*