[Reviews · NeurIPS 2020]

Review 1

Summary and Contributions: The paper proposes a framework for simultaneous pruning and robust training of deep nets. This is primarily accomplished by jointly optimizing a robust loss over the network parameters and a parameter mask. The proposed method is compared to some state of the art approaches on various datasets and architectures. The authors also present some evidence for the existence of robust sub-networks within pre-trained non-robust networks.

Strengths: As a disclaimer, none of the topics in this paper fall into my primary research area, though I have some familiarity. As such, I am unable to comment on the novelty of this work. The primary idea of the authors approach, namely joint optimization of a robust loss over the network parameters and a parameter mask, is quite intuitive. It seems likely the idea has been tried before, but the authors' main insight appears to be the application of the edge-popup algorithm of Ramanujan et al. 2020 in the context of a robust loss, followed by a fine-tuning step. While the idea is somewhat simple, the empirical results are convincing. Indeed, the primary content of the paper appears to be centered around the empirical results. From what I can tell, the experiments are well informed. Given the importance of the topic, I believe the empirical results are strong enough to warrant consideration. In particular, the evidence for the existence of robust sub-networks within non-robust trained networks seems complementary to the popular lottery ticket hypothesis and should be explored further in independent work.

Weaknesses: Given the calibur of the NeurIPS venue, it is usually preferable for accepted papers to provide new theoretical understanding, subtle/creative insight, or very in-depth analysis and interpretation of empirical results. The nature of this work is somewhat different, in that the main insight appears to be the straightforward combination of two existing ideas. However, the results are strong and could be of interest to the community.

Correctness: I am not aware of any mistakes in the methodology, motivation, or interpretation of the work.

Clarity: The paper is well written for the most part, though there are a number of grammatical errors. A thorough proof reading would substantially improve the presentation.

Relation to Prior Work: The authors clearly state their contributioons and sufficiently address prior work from my point of view, though, as I have said, I am not an expert in this area and cannot comment on the completeness of the relation to prior work.

Reproducibility: Yes

Additional Feedback: == After Author Response == I believe the author response was adequate in addressing the concerns of other reviewers regarding empirical evaluation and comparison to existing methods. The empirical results are good, but I stand by my original assessment that the work is a relatively incremental contribution in terms of the proposed objective and initialization. I keep my score as is.


Review 2

Summary and Contributions: This paper proposed a new pruning technique called HYDRA, that is able to compress networks with robustness preservation in the same time. And vast amount of experiments are conducted to evaluate their approach.

Strengths: The main idea of this paper is that the authors formulated pruning as an empirical risk minimization (ERM) problem and integrate it with a robust training objective. Another technique they employ an importance score based 
optimization approach to solve the ERM problem with a special scaled initialization of importance scores they proposed, which is quite interesting. They have conducted extensive experiments, with a supplementary material that contains very detailed experiment set up, ablation study, study of sparsity impact on robust training, increase in generalization with pruning, etc. The experiment section is quite impressive and significant!

Weaknesses: I am curious, why would the naive initialization of importance scores not work well? Why would SGD converge slower and worse with random initialization? This is unclear to me. What other initialization methods have been tried and why the proposed one work so well to ensure the performance is state-of-the-art? IIUC, the proposed initialization method is the major component to ensure performance gain. I am curious, what would be potential follow-up works and how to further close the performance gap between non-pruned and pruned network?

Correctness: yes

Clarity: yes

Relation to Prior Work: yes

Reproducibility: Yes

Additional Feedback: --After Rebuttal-- I thank the authors for addressing the comments and suggestions from reviewers. After reading other reviewers' responses and the author's rebuttal, I have decided to maintain the current score.


Review 3

Summary and Contributions: This paper presents a framework for reducing the number of the parameters while maintaining the robustness of a DNN model. The overall framework is based on a prior work, which presents an importance score to meaure the importance of different parameters. The author further notices that the initialization of the importance score is very important and presents a novel initialization strategy to improve the performance of the whole framework.

Strengths: 1. The presented initialization method is effective to achive faster convergence when the pruning ratio is high. 2. The author demonstrates the effctiveness of their method on different robust training methods.

Weaknesses: 1. The contribution of this paper is incremental. As the author claimed, the main contribution of this paper is a robust-training-objective-aware prunning technique. However, the main building block of this technique is based the prior work of importance score. The only improvement comes from the initilization strategy. Although, the author has empirically show its effectiveness, there lacks theoretical/intuative explainations about why it works. 2. The key idea of this paper is to incoperate the robust training into the prunning step. This shares the same spirit with ADMM-based prunning, thus I think the main comparison objective of this paper should be the Adv-ADMM. However, most of the results only contain the results of a weak baseline, i.e, Adv-LWM (e.g, Tabel3, Fig 1). It seems that Table 2 is the only place where show the comparison with Adv-ADMM. 3. There is no details about how to compute the verified robust accuracy in the experiment section.

Correctness: The claims are correct.

Clarity: This paper is well-written.

Relation to Prior Work: It is clearly discussed.

Reproducibility: Yes

Additional Feedback:


Review 4

Summary and Contributions: This paper proposed a method which makes pruning techniques aware of robust training objective, and evaluate the proposed method on different robust training objectives.

Strengths: This paper evaluates the proposed approach across different robust training objectives on a different dataset with multiple architectures and achieves better accuracy compared with some previous work.

Weaknesses: 1. The formulation of the problem seems straightforward, the authors should have more discussions on the reason why the proposed formulation can work better than previous work. weakness2. The paper should have more experiment results on the ImageNet dataset. For example, the author could compare the benign accuracy of compressed networks on ResNet-50 for ImageNet dataset with prior works.

Correctness: Claims and methods are correct.

Clarity: Yes

Relation to Prior Work: Yes

Reproducibility: Yes

Additional Feedback: I have read and appreciate the rebuttal. Most of my concerns has been addressed. This paper probably above the bar when taking all suggestions into account for the final version (as promised by the authors). I have increased my score to 6.


Review 5

Summary and Contributions: This paper focuses on 2 problems: (1) model compression using unstructured sparsity and (2) model robustness against various adversarial attacks. They improve an existing method (by initializing the importance weights to be same as the original weights and therefore biasing the importance towards magnitudes) and resulting networks bring better performance both in accuracy and in adversarial robustness.

Strengths: - Paper has a large set of experiments and it is clearly written.

Weaknesses: - It is not clear Hydra improves on adversarial attacks. It looks like test accuracy (benign) correlates with the adversarial accuracy (see Table:1). This is also observed by authors indirectly `L217: Our results confirm that the compressed networks show similar trend as non-compressed nets with these attacks`. It looks like as long as models are compressed properly the resulting models seem to be robust similar to dense networks. Therefore it is important to evaluate some SOTA sparse networks on compare them with HYDRA. It would be great to evaluate ResNet-50 checkpoints from (https://github.com/RAIVNLab/STR) and (https://github.com/google-research/rigl). - `L69: we develop a novel pruning technique, which is aware of the robust training objective, by formulating it as an empirical risk minimization problem, which we solve efficiently with SGD. -> It is not clear how the equation (2) is aware of the robust training objective as it seems like authors use {L_benign} in most of their pruning experiments (Section 4) Maybe I got it wrong, but couldn't see this being discussed. - `We focus on pruning after training, in particular, LWM based pruning, since it still outperforms multiple other techniques (Table 1 in Lee et al. [24]) and is a long-standing gold standard for pruning techniques` -> this table is on MNIST and no this is not a gold standard. There has been improvements over magnitude pruning since then (https://arxiv.org/abs/1710.01878) and see (https://arxiv.org/abs/1902.09574) for some better baselines. Authors only compare with LWM, which I believe, is not the best baseline to compare against and I recommend using wide-resnet or mobilenet instead of Vgg16 with 82% cifar-10 accuracy as it is not clear the observations on this setting (which is a bit irrelevant to the current sota) transfer to more recent architectures.

Correctness: This work is mostly experimental and their methodology seems appropriate.

Clarity: - This is first time I am reading `benign training`. (i.e. `pruning strategy that was developed for benign training, which per`). Similarly benign and robust accuracies. It would be nice to talk about them more generally at the beginning (i.e. `performance against adversarial attacks) before defining each later in the text. - Maybe I missed, but it is not clear which loss is used in experiments at Section 4.1. Is it L_ver or L_benign? In Algorithm 1 there is also L_pretrain defined, is this different than L_finetune? It would be nice to give these details concisely. - In Table:1 does different initialization corresponds to Hydra with different importance initialization? If so, it would be nice to group them with scaled initialization, as currently they sound look like a version of `scratch` training. - L215: networks in not arising -> is not

Relation to Prior Work: - Training from scratch doesn't work well and reported/discussed by various work [1, 2, 3]. It would be nice to cite them. [1] https://arxiv.org/abs/1803.03635 [2] https://arxiv.org/pdf/1810.02340 [3] https://arxiv.org/abs/1810.05270 [4] https://arxiv.org/abs/1906.10732

Reproducibility: Yes

Additional Feedback: - In abstract: `We also demonstrate the existence of highly robust sub-networks within non-robust networks.` This feels a bit out of blue. Maybe it worth leaving out from the abstract or motivating how it connects to the story. Similarly in the main text, why this is a relevant observation? I would see the utility if we can find these networks without training, but when we do training this is pretty much like doing pruning. - For `Adv-ADMM`, do you have ResNet experiments? - two key challenges: lack of robustness against adversarial attacks and large neural network size -> (for the same benign performance) Is it clear that large networks bring safety? Or smaller network means more safety? Or are they orthogonal? I am willing to update my score, if authors can address my concerns about experimental validation. --After Rebuttal-- I thank authors for their response and the clarifications. It looks like I had few misunderstandings. I updated my score. I still encourage authors to isolate the effect of pruning algorithm from the loss optimized (i.e. the adversarial loss) more clearly. It feels like there might be an unfair comparison to compare algorithms pruned with regular loss and Hydra (which is pruned with adversarial loss). I encourage authors to make this clear and ideally re-run other baselines with the adversarial loss.

[Author Response · NeurIPS 2020]

We thank all reviewers for their thoughtful and constructive feedback. We are encouraged that the reviewers find our
idea of making pruning aware of robust training objectives intuitive (**R1**), our experiments well-informed (**R1**) and
extensive (**R2**, **R5**), and success of our method, across different robust training objectives, datasets, architectures, and
pruning ratios, an accomplishment (**R2**, **R3**, **R4**). We are pleased that **R1** finds our observation on the existence of
robust sub-networks timely and that **R2** finds our extensive supplementary material impressive.

Two major concerns were 1) additional insights on why proposed initialization is highly effective (**R2**, **R3**, **R4**) and 2)
additional comparison with Adv-ADMM baseline (**R3**, **R5**). We address both, along with other comments, below and
will incorporate all feedback in the updated version of the paper.

[**R2**, **R3**, **R4**] **Why is the proposed initialization effective? Why do random initializations not work so well?**
We show in Fig. 1 (left) that with proposed initialization SGD converges faster and to a better pruned network, in
comparison to widely used random initializations. This is because with our initialization SGD enjoys much higher
magnitude gradients throughout the optimization in the pruning step (Fig. 1 right). We will add in-depth analysis of it
in the main paper. [**R2**] **What other initializations were tried?** We now compare with two more initializations, based
on Dirac delta function and orthogonal matrices, along with four other widely used initializations.

[**R3**, **R5**] **More comparison with Adv-ADMM:** Following reviewers' suggestion, we now provide a comparison along
six more recent architectures (Table 1). Our method achieves better accuracy and robustness, simultaneously, across all
of them. Furthermore, when Adv-ADMM fails to even converge for MobileNet, a highly compact network, we achieve
non-trivial performance. We already provide comparison at different pruning ratios in Table 2 of the main paper. [**R3**]
**Verified robustness?** We use existing techniques IBP, Mixtrain, randomized smoothing to compute verified robustness.

Figure 1: We compare proposed initialization with six other widely used initializations. With proposed initialization in the pruning step, SGD converges faster and to a better architecture (left), since it enjoys higher magnitude gradients throughout (right). (CIFAR10, 99% pruning)

| Network | Adv-ADMM | Ours | Δ |
|---|---|---|---|
| ResNet-18 | 58.7/36.1 | 69.0/41.6 | +10.3/+5.5 |
| ResNet-34 | 68.8/41.5 | 71.8/44.4 | +3.0/+2.9 |
| ResNet-50 | 69.1/42.2 | 73.9/45.3 | +4.8/+3.1 |
| WRN-28-2 | 48.3/30.9 | 54.2/34.1 | +5.9/+3.2 |
| GoogleNet | 53.4/33.8 | 66.7/40.1 | +13.3/+6.3 |
| MobileNet-v2 | 10.0/10.0 | 39.7/26.4 | +29.7/+16.4 |

Table 1: Comparing test accuracy/robustness (*era*) with Adv-ADMM (CIFAR10 dataset, 99% pruning). Our approach outperforms Adv-ADMM across all network architectures.

[**R5**] **Comparison with other pruning strategies:** Following reviewer's suggestion, we now provide comparison with
techniques from each of three pruning paradigms, i.e., pruning before training (SNIP), pruning with training (STR), and
pruning followed by re-training (Adv-LWM, Adv-ADMM – these were already covered in the paper). We find that it is
not always the case that an existing pruning strategy will also be successful with robust training. For example, SNIP
performs equivalent to random pruning, i.e., scratch in Table-1 of main paper, when tested with adversarial training (*era*
for SNIP=27.2%, scratch=24.6% at 99% pruning ratio). Similarly, STR with adversarial training achieves only 33.2%
*era*, where our approach achieves 41.6% *era* at 99% pruning ratio for the ResNet-18 network and CIFAR10 dataset.

[**R5**] **Better architectures like Wide-ResNet, MobileNet:** We already use Wide-ResNet-28-4 in major experiments
(Table-3 in the paper). Following reviewer's suggestion, we demonstrate the success of our method with recent
architectures in Table 1. We achieve SOTA robustness in the context of highly compressed networks across multiple
architectures. [**R5**] **Authors use $L_{benign}$ in most of their pruning experiments:** We think that this comment is a
misunderstanding as we use $L_{adv}$ or $L_{ver}$ in most experiments (L146). We thank the reviewer for providing additional
suggestions on related work and clarity, which we will incorporate.

[**R5**] **As long as models are compressed properly, resulting models are robust . . . compare with RIGL, STR:** We
find that it is critical for the pruning step to be aware of the robustness objective. However, techniques like RIGL/STR,
don't account for the robustness objective while pruning. As suggested by the reviewer, we experimented with their
pre-trained checkpoints and observed 0-0.5% robust accuracy. This further validates the importance of our approach.

[**R4**] **More experiments on ImageNet:** Following reviewer's suggestion, we also demonstrate success with provable
robustness on ImageNet. Our approach achieves 47% provable robustness, while pre-trained nets has 49% and Adv-
LWM achieves 44%, at $||\epsilon||_2 = 0.5$, 90% pruning, using randomized smoothing. Note that this is the *first* work to
demonstrate the success of pruning with robust training on the scale of ImageNet. Earlier works [14, 44] only perform
experiments with CIFAR10/SVHN. We hope that the reviewer sees this as a *strength*, not a weakness of our paper.

[**R1**] **Combination of two existing ideas?** We clarify that this combination is in itself novel, and that our proposed
initialization is a key driver for success. This insight allows us to achieve SOTA accuracy and robustness, for compressed
networks, across different pruning ratios, architectures, and datasets (including provable robustness at ImageNet scale).

[Meta-Review · NeurIPS 2020]

The paper presents a method to prune a network while making it more adversarially robust. The idea is simple and has thorough experimental validation. The reviewers suggest several additional experiments, especially using RN50 on ImageNet which would further strengthen the paper. Additionally, the strongest possible baseline should be used throughout.